# Auto-Denoising for EEG Signals Using Generative Adversarial Network

**DOI:** 10.3390/s22051750

**Published:** 2022-02-23

**Authors:** Yang An, Hak Keung Lam, Sai Ho Ling

**Affiliations:** 1School of Electrical and Data Engineering, University of Technology Sydney, Ultimo, NSW 2007, Australia; yang.an-1@student.uts.edu.au; 2Department of Engineering, King’s College London, London WC2R 2LS, UK; hak-keung.lam@kcl.ac.uk

**Keywords:** brain–computer interface, electroencephalogram, convolutional neural network, generative adversarial network, denoising, normalization

## Abstract

The brain–computer interface (BCI) has many applications in various fields. In EEG-based research, an essential step is signal denoising. In this paper, a generative adversarial network (GAN)-based denoising method is proposed to denoise the multichannel EEG signal automatically. A new loss function is defined to ensure that the filtered signal can retain as much effective original information and energy as possible. This model can imitate and integrate artificial denoising methods, which reduces processing time; hence it can be used for a large amount of data processing. Compared to other neural network denoising models, the proposed model has one more discriminator, which always judges whether the noise is filtered out. The generator is constantly changing the denoising way. To ensure the GAN model generates EEG signals stably, a new normalization method called sample entropy threshold and energy threshold-based (SETET) normalization is proposed to check the abnormal signals and limit the range of EEG signals. After the denoising system is established, although the denoising model uses the different subjects’ data for training, it can still apply to the new subjects’ data denoising. The experiments discussed in this paper employ the HaLT public dataset. Correlation and root mean square error (RMSE) are used as evaluation criteria. Results reveal that the proposed automatic GAN denoising network achieves the same performance as the manual hybrid artificial denoising method. Moreover, the GAN network makes the denoising process automatic, representing a significant reduction in time.

## 1. Introduction

An electroencephalogram (EEG) is a brain signal generated by a non-implantable brain–computer interface (BCI). The EEG signal can reflect a variety of brain activities obtained by attaching electrodes to the scalp. Therefore, the EEG signals can be easily affected by unexpected noise such as eye blinking, heartbeat etc. These noisy signals may generate higher energy than the original EEG signal, making EEG-based research difficult. Thus, denoising is an important area of EEG research.

Head muscle movements, blinking, eye movements heartbeat are the common factors to generate the noise. Traditionally, the commonly used denoising methods are wavelet transform (WT) denoising, independent component analysis (ICA) denoising, and empirical mode decomposition (EMD) denoising. References [1,2] used wavelet denoising based on soft or hard threshold selection. First, the signal was normalized, and then the signal was decomposed by a wavelet. The researchers used soft and hard thresholds to remove the noise part from the decomposed sub-signals and the filtered signal was reconstructed. Compared to [1], Reference [2] used different threshold selection methods and different wavelets to decompose the signal, which were then tested separately using different noises. In addition, an adaptive filtering method was employed, which effectively removed the influence of noise, such as that of heartbeat [3]. The authors also compared this method with the wavelet denoising method and obtained good results.

Like the wavelet denoising method, EMD is a signal decomposition method. Reference [4] used the EMD filtering method; however, this method can only decompose the signal fixedly which is not as flexible as the wavelet method. Accordingly, the use of EMD denoising has certain limitations. In addition to these two decomposition methods, the analysis of signal components on multichannel signals serves as an effective denoising method. The methods employed include ICA and PCA. Reference [5] used PCA to reduce the dimensions of multi-dimensional EEG signals and then used the density estimation blind source separation algorithm to denoise the signal. The authors first reduced the dimension and then analyzed whether each component was a noise component through calculating the correlation coefficients and other evaluating parameters. After this, they reconstructed the remaining components to achieve noise reduction. This method also has limitations in that the EEG signal must have an EOG or other noise reference. The similar methods also include canonical correlation analysis (CCA) denoising and some muscle artifact removal methods [6,7,8,9].

In addition to these traditional denoising methods, there are denoising methods based on neural networks [10,11]. Reference [10] proposed a Schrodinger wave equation based on alternative neural information processing architecture, extracting effective motor imagery features while removing EEG noise. Reference [11] proposed a growing artificial neural network using the simultaneous perturbation method. This network can optimize the number of nodes in the hidden layer while denoising the signal. Reference [12] used a one-dimensional residual network for noise reduction. The author first divided the network into three parts to extract the features separately. Then they combined the features as the output and used the loss function to minimize the square error of inputs and outputs.

Recently, the generative adversarial network (GAN) was introduced to remove the noise from signals. Reference [13] proposed a speech enhancement generative model. The authors applied the model to voice signals and tested the model using 30 subjects’ data within 40 types of noise. They obtained better performance compared to some state-of-the-art methods. Reference [14] also introduced a generative model to remove the noise from the ECG signal. They added the least square term into the generative model, which can tackle the issue of vanishing gradients. All of these methods are signal denoising. The generative model also can apply to image denoising. References [15,16] used Wasserstein GAN, GAN, and f-GAN to process CT images. Reference [17] applied GAN to repair the images and [18] used GAN to construct images. In addition, GAN improved the quality of images and removed noise from images [19,20,21]. These applications all belong to image filtering. EEG data can also be seen as an image by high-dimensional data. Compared to some traditional filtering networks, the GAN network has an extra discriminator. This extra discriminator can help the network to determine whether the data were successfully filtered. Therefore, the EEG signals can also be denoised using GAN.

From the literature review, there is a problem that we need to overcome. Denoising is a necessary step. Although most of these denoising methods can achieve a good performance, they still have limitations. ICA is the denoising method with the best performance compared to other traditional denoising methods. However, ICA requires manual judgment and a selection of noise components one by one, which belongs to artificial denoising. Thus, this method is time-consuming and not automatic. In addition, participants’ brain waves are different. Some deep learning-based denoising methods use fixed subjects’ data for training; however, the performance is significantly reduced when the model is applied to other subjects’ data. To overcome these problems, we propose a GAN-based denoising method that can reduce the computation time and makes the whole process perform automatically. Moreover, it performs well when the testing data of subjects are different from the training subjects. After the model is trained, when new subject data are acquired using the same device, the model can still denoise the signal. Although the new signal was never involved in the training of the model, the model could still extract the noisy features of the EEG signal and remove the noise.

Another problem is that it is challenging to generate EEG signals by using the GAN model. When using GAN to generate images, the range of the image data is normally between 0 and 1, while the range of the EEG signal is uncertain. Some channel signal amplitude values are higher, above 50 μV, but some are lower, between −10 μV and 10 μV. Sometimes, when humans blink their eyes, move their facial muscles, or when other effects occur, the amplitude value can suddenly extend into the hundreds of micro volts. Thus, if we do not do any normalization, the range of the new EEG obtained is not controllable, meaning that the GAN is also unstable. Although the wave shapes of some signals generated by GAN are similar to the target signal without any normalization, it can still lose some signal information. Therefore, it is very important to control the range of input EEG data before using the GAN model. We propose a sample entropy threshold and an energy threshold-based normalization method to overcome these problems. This method can check the abnormal parts of the signal and limit the range of the EEG signals, which is particularly helpful in relation to the GAN generating EEG signals.

In summary, there are two significant contributions to this paper. Firstly, sample entropy and an energy threshold-based data normalization method are introduced. The purpose is to apply the idea of image restoration to EEG signal denoising. However, EEG differs from image filtering in that the amplitude of the EEG signal has no fixed range. The GAN model may generate abnormal signals, such as a signal with a strange shape that already loses the original EEG characteristics. Thus, before using the GAN model, the EEG signal should be normalized. Using the proposed normalization method, the GAN model can stably generate the EEG signal because the value of EEG data is limited to a small range, which can be seen as some images. Compared to other normalization functions, such as tanh function, the proposed method can avoid information loss because part of noisy EEG signals are selected and processed before inputting into the generative model. Thus, during the model training process, the performance is not influenced by the mutation signals. The proposed normalization method is thereby helpful in terms of the GAN model stably generating EEG signals.

Secondly, the GAN-based blind denoising method for removing noise is proposed. The GAN model can replace artificial denoising and still obtain the same filtering performance. Compared to the artificial denoising method, our model is automatic, which reduces the computation time. In addition, when our model uses the different subjects’ data for training, it can still apply to the different subjects’ data denoising. The GAN model has an extra discriminative dimension compared to the deep network denoising model. The discriminator always judges whether the noise is filtered out. The generator is constantly changing the denoising way. During this process, the discriminator and generator update their parameters to improve the filtering performance.

## 2. Methodology

The proposed GAN-based denoising system is shown in Figure 1. There are three steps:
The EEG data are pre-processed, including frequency filtering, re-reference, and baseline removal;The processed data are normalized to a range. This step uses energy threshold and entropy threshold to select the noisy part of the EEG data;The original EEG data are input into the trained generator of the GAN model to obtain the cleaned EEG data.

### 2.1. Data Pre-Processing and Normalization

The raw EEG data are filtered by 1 to 40 Hz because the main features of EEG exist in this frequency band [22]. This can also remove some high-frequency noise, such as 50 Hz or 60 Hz linear noise. Re-reference involves using the raw EEG data to minus the mean value of all the channels. The function of this step is to correct the reference electrode close to zero.

Next, we develop the sample entropy threshold and energy threshold-based normalization method, as shown in Figure 2. We aim to normalize the original EEG signal to limit the range of the signal. If the subject blinks, the channel’s signal near the eye can experience a sudden and significant fluctuation. However, if we normalize the data by using the signal to divide the max value, it can result in the overall energy being particularly small. Thus, we need to find a way to obtain the noisy position of the original signal and process the bad parts before we normalize the data.

First, sample entropy detection is used [23]. Entropy is a measure of signal confusion. If a signal is abnormal, the sample entropy of this signal should be minimal. Thus, we define a sample entropy threshold *k*. When the signal sample entropy is greater than *k*, the signal is defined as a normal signal. When the entropy value of the signal sample is smaller than *k*, there may be a sudden change in the signal, and the signal will need to be processed. The value of *k* is selected based on experience. This threshold is used to check whether the signal fluctuates sharply and the value selection is not sensitive. It is related to the signal length. Usually, for EEG signals, the range of sample entropy is from 0 to 1.5. When the signal is seriously affected by eye movement, the sample entropy should be very small because it does not contain much EEG information; however, it only has a significant fluctuation. Generally, sample entropy less than 0.2 means that the data have a lot of information loss.

We can manually select some data that are slightly affected by eye movement to calculate the mean sample entropy $SE_{a}$ by using all channels’ data. Select some front channels’ signals to calculate the mean sample entropy $SE_{f}$. Then, select some central channels’ signals to calculate the mean sample entropy $SE_{c}$. The data of the channels close to the eye may be significantly affected by eye blinking; therefore, the mean sample entropy should be small. The data in the central area of the brain are ordinarily stable; therefore, the sample entropy should be large. If there is not much difference among the three sample entropy, $SE_{f}$ can be used as the threshold *k*. If the three entropy is quite different, a value can be selected between $SE_{a}$ and $SE_{c}$ as the threshold *k*.

Next, we calculate the entropy of the signal for each channel and select the signal that needs to be processed. The noise of the signal may appear at a certain time point or a short period. Therefore, we first divide the signal into *n* segments. For each segment of the signal, the mean energy is calculated. We use the *L*1 norm of the signal as the energy. The energy of the sudden change signal should be higher; therefore, we use the same means to set an energy-based threshold *e* to select the signal segment that needs to be processed. When the mean energy is less than *e*, the signal is normal. The segment may contain sudden fluctuation when the energy is greater than *e*. When calculating the normalization coefficient, this part should be ignored. Usually, the energy of the EEG signal should be below 30. When the artefact noise comes, the mean energy should be over 50. Thus, 50 can be used as a reference of threshold *e*, which is applicable for most EEG signals.

Finally, we calculate the absolute max value of normal segments of data as the normalization coefficient *m*. Then, the whole data divides *m* to achieve the purpose of normalization:(1)$$S=\frac{X}{m}$$
where X is original data and S is the normalized data. We record the max value of each group of data and use the parameter *m* to reconstruct the original signal range after generating a new signal. After normalization, the processed signal into the GAN network is employed to train the model.

The selected threshold is not related to the subjects because the selection of the threshold depends on the mean sample entropy. For different subjects, the mean sample entropy is also different. Thus, the threshold is not fixed, which changes dynamically according to the different subjects.

The purpose of the threshold is to select the affected data segment and then pick it out for processing. If the signal is abnormal, the calculated sample entropy would be very small. We assume that the threshold is set as the smallest threshold $SE_{a}$. Any value smaller than this value must be an abnormal signal segment. For another situation, we assume that we set the threshold as the largest threshold $SE_{c}$. Although some normal data could also be filtered out, these data can still be filtered back by the energy threshold in the next step.

For a worst situation, if both thresholds are set too large and the normal data are filtered out, then these data will be divided by a small value during normalization. Finally, the signal will still not lose too much information. Thus, the threshold is not very sensitive within the selection range and the selection range is determined automatically based on the mean sample entropy of the EEG signal.

### 2.2. Construct Generative Adversarial Network

In this paper, the generative adversarial network (GAN) is used to denoise the EEG signal. There are two parts—the generator and the discriminator. For the generator, the input is an unfiltered signal and the output is a clean signal. Therefore, we need to obtain some clean signals (e.g., 30% for each subject) to serve as the training data before using this network. Thus, we take some of the data to perform artificial filtering. The filtering method is mainly based on the ICA. We also use the wavelet denoising method [1] and EMD denoising method [4]. Firstly, we use the ICLabel technique to process the raw signal to obtain the estimated probability of each noise. We select several kinds of noise with high probability for further analysis. We use the EEGLAB toolkit [24] to plot the spectrum of each independent component and the distribution of brain energy. Then, we determine whether this independent component is a noise component based on the signal waveform, spectrum and energy distribution. After that, the noise component is removed to obtain the filtered signal. After artificial denoising by ICA, the EMD [4] and wavelet transform filtering methods [1] are used for secondary filtering. Eventually, a relatively clean EEG signal is obtained. Thus, the denoising method can be seen as a hybrid artificial method.

#### 2.2.1. Build Generator and Discriminator Structures

In the network, the generator first compresses the original multi-dimensional signal and then extracts the main features of the signal. After that, it reconstructs the signal according to the feature vector. Therefore, aside from the generator, we also need to define a discriminator. The discriminator is used to determine whether the signal generated is a clean EEG signal. Training the model is to find the Nash equilibrium where the generator can generate clean signals and the discriminator can classify the noisy signals and clean signals. The designed generator and discriminator network structures are shown in Figure 3 and Figure 4, respectively. The suggested parameters are tabulated in Table 1 and Table 2.

#### 2.2.2. GAN Model Training

The training process is shown in Figure 5 and Figure 6. Referring to Figure 5, the noisy EEG data *S* are input into the generator, and the output are the generated EEG data $G\left(S\right)$. The generated data can be denoised data or they can also be the data still with noise. Next, the generated data $G\left(S\right)$ and the clean data *y* are input into the discriminator. The purpose is to make the discriminator have ability to classify noisy and clean data. Next, the output of the generator is the input of the discriminator can produce the outputs $D\left(y\right)$ and $D\left(G\left(S\right)\right)$. They can be regarded as the variables that reflect whether the output of the generator are clean EEG data or not. The discriminator can be seen as a binary classification classifier. The loss $L_{D}$ of the discriminator depends on these two outputs. If the output value is large which means that the data are clean; otherwise, the data are noisy.

For the generator (shown in Figure 6), the loss $L_{G}$ depends on the feedback from the discriminator. We assume that the discriminator can accurately classify the clean data *y* and generated data $G\left(S\right)$. Then, the purpose of the generator is to generate more clean data to make the output value $D\left(G\left(S\right)\right)$ from the discriminator become very large. At the same time, the mean square error between the clean data *y* and the generated data $G\left(S\right)$ should also be considered as another updating factor because the generator should also ensure that the generated data $G\left(S\right)$ cannot lose too much original data information. Therefore, the mean square error and the output of the discriminator are jointly used to calculate the loss $L_{G}$ of the generator.

The output value of the discriminator can be regarded as the degree of denoising level. When the clean data are input into the discriminator, the output value is expected to be large. When the noisy data are input into the discriminator, the output value is expected to be small, which means that the discriminator has the ability to classify the clean data and noisy data. When the input of the discriminator is from the generator, a large output value of the discriminator is returned, which implies that the generated data are clean. Otherwise, the generated data are noisy when a small value of the output of the discriminator is returned. Based on this condition, this value will become large when the generator is well trained. It means that the trained generator has the ability to remove the noise. Thus, for the generator, the output value from the discriminator is expected to be as large as possible if the noise is successfully removed.

Referring to Equation (3), the loss function of generator should have two loss items. The first item is the mean output of discriminator *D* when inputting the target data *y*. The purpose is that when the generator’s output is input into the discriminator, it is best if the value of the discriminator output is as large as possible. The second item is the mean output of discriminator *D* when the generative data $G\left(S\right)$ are used as input. The purpose is that the contribution of a signal output by the generator should be as close as possible to the contribution of the artificially filtered signal.

Referring to Equation (2), the loss function of discriminator should also have two loss items. The first item is the mean output of discriminator *D* when the generative data $G\left(S\right)$ are used as input. The discriminator should have the ability to distinguish whether the signal is an artificially filtered signal or a generated signal. When the artificially filtered signal comes, the output of the discriminator should be as large as possible. When the generated filtered signal comes, the output of the discriminator should be as small as possible. The second item is the mean square error between the generated signal $G\left(S\right)$ and target signal *y*. When the model is used to filter the signal, the filtered signal should keep as much original information as possible.

The traditional GAN network uses Jensen–Shannon (JS) divergence to measure the distance between two distributions [25,26]. Multi-dimensional signals are the same as images. They are all high-dimensional data; therefore, the target distribution and the real distribution are difficult to overlap in space, and they are difficult to train using the traditional GAN. Therefore, the generative model in this paper is based on the Wasserstein generative adversarial network (WGAN). The distance between the two distributions is measured by the earth-moving distance [27]. Then, we combine the conditions mentioned above with the loss of WGAN. The final loss function is defined in Equations (2) and (3):(2)$$L_{D}=-\frac{1}{N}\overset{N}{\underset{i=1}{\sum }}\left[D\left(y_{i}\right)\right]+\frac{1}{N}\overset{N}{\underset{i=1}{\sum }}\left[D\left(G\left(S_{i}\right)\right)\right]$$
(3)$$L_{G}=-\alpha \frac{1}{N}\overset{N}{\underset{i=1}{\sum }}\left[D\left(G\left(S_{i}\right)\right)\right]+\beta \frac{1}{N}\overset{N}{\underset{i=1}{\sum }}\left[{\left(y_{i}-G\left(S_{i}\right)\right)}^{2}\right]$$
where α and β are used to balance the loss of the generator. Because the GAN model is composed of two networks, the learning process of the two networks should be balanced. In other words, the ability of the discriminator to determine whether it is noise can also not be too good or too bad. Thus, the two balanced networks can improve their performance together. The ability of the generator reaches its limit if both mean square errors and the output of the discriminator are almost unchanged. If we continue to train the model, the performance of the discriminator may change, resulting in the performance of the two networks being out of balance. This, in turn, results in the poor performance of the generator. Therefore, we should stop training the network when both the mean square error and the output of the discriminator no longer change or decrease slightly.

We calculate the mean square error between the output of the generator and the clean EEG data in each iteration during the updating process:(4)$$MSE\left(U,V\right)=\frac{1}{K}\overset{K}{\underset{k=1}{\sum }}\frac{\sum_{c=1}^{N}{\left(U_{kc}-V_{kc}\right)}^{2}}{N}$$
where $U_{kc}$ is the $k_{th}$ trial $c_{th}$ channel of clean data. $V_{kc}$ is output of the generator. *N* is the total channel number in each trial. *K* is the total trial number. $MSE\left(U,V\right)$ is the mean square error of *U* and *V*.

If the error does not decrease anymore which means that the network cannot gain its denoising ability. As a result, the trained generator can be used as a noise filter. The processed and normalized EEG signal are fed into the generator to perform the signal denoising operation.

## 3. Experiments and Results

### 3.1. Dataset

In this study, the HaLT dataset is used to demonstrate the performance of the proposed denoising method, which was published in 2018 [28]. There are 12 subjects and the trial number of each participant in each category is about 160. There is a total of approximately 950 trials for each participant with six imaginary motion states. There are 22 channels of data. However, channels 11 and 12 are reference channels while channel 22 is the pulse signal channel; thus, there are only 19 valid data channels. Participants make corresponding imaginary behaviors according to the random prompts on the screen. The sampling frequency of this dataset is 200 Hz, and in each experiment, it only collects data for about one second. To facilitate the experiment, we cut the data of each experiment to a length of 200. The data for this period are motor imagery data which are obtained when the subject sees the instructions. These data are pure motor imagery data, excluding the rest data and preparation data. Motor imagery tasks include left hand, right hand, left leg, right leg, and tongue imagery movement. The number of trials for each subject in the HaLT dataset is tabulated in Table 3.

### 3.2. Experiment

There are two groups of experiments. The first experiment aims to denoise the data using the network which is trained by single-subject data. The second experiment aims to denoise the data using the network which is trained by multiple subject data. The experiment is mainly based on the HaLT dataset. For the HaLT dataset, the data size is 19 × 200. The channel number is 19, and the data length is 200. The dataset has a total of 12 subjects. In the experiment, the sample entropy of the signal with sudden and large fluctuations is below 0.2, and the sample entropy of the signal with slightly abnormal fluctuations is between 0.2 and 0.3. Thus, the sample entropy threshold *k* is set to 0.3. When the sudden change signal occurs, the energy should be above 50. The energy of the normal signal is between 20 and 30. Hence, the energy threshold *e* is set to 50. The processed EEG data are normalized. The optimizer is a root mean squared propagation (RMSProp). The learning rates *ƞ* of both networks are 0.0002. In the loss function, the balance parameter α is set at 0.1 and β is set at 1.

#### 3.2.1. Denoise Using Single-Subject EEG Data

We perform the GAN denoising method on the HaLT dataset. Firstly, we artificially denoise 30% of the data of every single subject and then use the original signal as input. Furthermore, 70% of the data of every single subject are used as a testing set. The artificially denoised signal is used as a real sample to train the generator and discriminator. After 500 epochs of training, the GAN model is obtained. The test sample is then input into the generator.

We use two evaluation methods. The first is the mean correlation. The filtered signal should be as close to the original signal as possible because the filtered signal should not lose the original information of the signal as much as possible. The higher the similarity, the more information the filtered signal retains from the original signal. The second evaluation method is to calculate the RMSE. Compared to the signal before filtering, the filtered signal will lose noise signals. Thus, the energy may decrease; however, it should not be by too much because the filtered data should retain as much original information as possible. When the correlation is higher and the RMSE is lower, the filtering performance should be better. The mean correlation and the root mean square error are shown in Equations (5) and (6) below.
(5)$$R\left(X,X\prime \right)=\frac{1}{K}\overset{K}{\underset{k=1}{\sum }}\frac{1}{N}\overset{N}{\underset{c=1}{\sum }}\frac{\left(X_{kc}-\bar{X_{kc}}\right)\left(X_{kc}\prime -\bar{X_{kc}\prime }\right)}{\sqrt{{\left(\sigma_{X_{kc}}\right)}^{2}{\left(\sigma_{X\prime_{kc}}\right)}^{2}}}$$
where *X* is the original signal and *X*′ is the denoised signal. $X_{kci}$ is the $k_{th}$ trial, $c_{th}$ channel $i_{th}$ point of signal *X*. σ is the variance. *K* is the total number of trials. *N* is the total channel number in each trial. The second evaluation measures the RMSE.
(6)$$RMSE\left(X,X\prime \right)=\frac{1}{K}\overset{K}{\underset{k=1}{\sum }}\sqrt{\frac{\sum_{c=1}^{N}{\left(X_{kc}-X\prime_{kc}\right)}^{2}}{N}}$$
where $X_{ki}$ is the $k_{th}$ trial, $c_{th}$ channel of signal *X*. *K* is the total number of trials; *N* is the total channel number in each trial. $RMSE\left(X,X\prime \right)$ is the root mean square error of *X* and *X′*. The results are tabulated in Table 4.

There are some graph results, which are shown in Figure 7, Figure 8 and Figure 9. They can be divided into three groups: the EEG data that are slightly affected, the EEG data that are moderately affected, and the EEG data that are seriously affected by noise signals. The top side of the figure is the original EEG signal while the bottom is the result after filtering.

#### 3.2.2. Denoise Using Multiple Subject EEG Data

We merge five subjects’ data and select 30% of the data as training data to train the GAN model. Then, we use the remaining data of these five subjects and the other seven subjects’ data as the test set to test the denoising network separately. The five subjects are A, B, C, G, and H because these subjects contain all the types of noisy data, including the EEG data slightly affected by noise, the data moderately affected by noise, and the data seriously affected by noise. Therefore, in this process, the network can learn more features of the different types of noise. The purpose of this experiment is to prove that the GAN model can be used not simply on the EEG data which belong to the training subjects, but also on the data which belong to the subjects that have never been used before. In this experiment, the training data pertains to the five subjects. The remaining data of the five subjects and the other seven subjects are the EEG data which have not been used in the network before. The results are tabulated in Table 5.

## 4. Discussion

The first experiment uses single-subject data to train the model and test the model using each subject’s data. From Table 4, it can be seen that most of the noise is successfully filtered out. By observing two evaluation indicators, most of the filtered signal can maintain most of the original information. However, some signals do not get good filtered results. For example, the similarity of subject C is worse. This is because the data of subject C has a large amount of data that is seriously affected. Some channels are close to the ears and eyes. The amplitude change in these channels of data is volatile. Therefore, the RMSE between the filtered and original data is relatively large.

The second experiment employs multiple subjects’ data to train and test the model using each subject’s data. From Table 5, merging all the data is better than the testing data for each participant, especially for subjects C and G. Because the training samples of the data are increased, these data samples contain different types of noise signals from different subjects. There are more noisy signals, which allows the network to learn different forms of denoising. Therefore, in practical applications, it can select a variety of noise signals from different participants to train the generative model so that the trained network can be used to various denoise types of EEG signals.

We are more concerned about whether the denoised signal can still retain the original data information during the denoising step. We can analyze the data from the time and frequency domain. The RMSE is used to measure the degree of information retention of the original signal in the time domain. We have already evaluated the data by this indicator in previous experiments. From the results, little time information is lost. Thus, we continue to analyze the data from the frequency domain. The data are a motor imagery signal. The useful frequency information should exist in alpha and beta bands. Thus, we focus on whether the denoised signal retains the original frequency information in the 5–25 Hz band.

There are four main types of noise in EEG signals: eye movement artifact noise, EMG noise, heartbeat noise, and power line noise. Eye movement noise is in the low-frequency range, normally 1–4 Hz. It appears as a sudden fluctuation in a short period. The EMG noise is a signal over 30 Hz within a significant fluctuation. Power line noise is a 50 Hz sinusoidal signal. Since the signal is filtered at 1–40 Hz in the pre-processing stage, we do not need to consider this type of noise. Heartbeat noise is difficult to accurately distinguish in the frequency domain; however, the heartbeat is a regular noise so that it can be easily determined in the time domain. However, the data used in this paper are short time data; therefore, the heartbeat noise can also be ignored.

Figure 10 presents a group of signals with eye movement artifact noise in the time domain. From the figure, the signals have significant sudden fluctuation in the front channels. After denoising, we can see that these sudden fluctuations are removed in Figure 11. The shape of the signal in other channels is almost unchanged in the time domain. We also draw the mean frequency spectrum of these front channels in Figure 12. We compare the frequency spectrum of original data and denoised data. It appears that the energy in the low-frequency band is decreased. The power in the other frequency band is almost unchanged. Thus, our model can remove the eye movement artifact noise and also retain the most original data information.

Figure 13 presents a group of signals with EMG noise in the time domain. From the figure, the signals contain high-frequency fluctuation. After denoising, we can see that these high-frequency fluctuations are removed in Figure 14. We also draw the mean frequency spectrum of motor imagery-related channels in Figure 15. We compare the frequency spectrum of original data and denoised data. It appears that the energy over 30 Hz is decreased. In other frequency bands, the signals may be slightly affected; however, most of the frequency information is retained. However, we cannot ensure that the EMG noise can be accurately and completely removed because we do not have ground truth. We can only say that this denoising method can be used to reduce the EMG noise and retain the most original information.

Our purpose is to use the GAN model to replace the artificial denoising method. Thus, we also compare the GAN-based denoising method with the artificial denoising method. The correlation and RMSE results are tabulated in Table 6.

From the results, it can be seen that most of the GAN filtering effects are close to those of the artificial filtering performance, and some GAN filtering effects are better than the artificial filtering results. GAN is better than artificial filtering in terms of time and labor costs because the GAN model is an automatic method. Therefore, the proposed GAN network can reduce the processing time. Artificial filtering can be seen as combining ICA with manual selection by human eyes, threshold-based EMD, and wavelet methods. Thus, the generative model can also explain how to combine multiple effective filtering methods.

Compared to other networks, GAN has both a generator and a discriminator. Thus, it can always judge whether the generated signal is still noisy. If there is noise, the model will keep training. While other models cannot judge whether the generated signal is clean or not, they just simply output the filtered signal. We apply GAN to repair the multi-dimensional EEG signal. A new loss function is defined specifically for EEG filtering, which ensures that the filtered signal can retain effective information and energy. Since EEG is a multi-dimensional signal and the signal is unstable, the normalization method is used to process the EEG signal before inputting that into the network. The processed EEG signal can allow the GAN model to better extract the EEG signal features. Once the denoising system is established, the model can still denoise the signal even though the new subject’s signal was never involved in the training of the model.

Due to a large amount of data, there is no guarantee that the artificial method can always filter the data very well because humans need to manually select the destructive components according to personal experience. However, there are no fixed evaluation criteria to remove the destructive components. It can also be seen from the results that some artificial filtering results are not very good. Therefore, denoise performance can be improved if a cleaner signal is used as the learning target.

By using traditional ICA, we need to manually filter each subject, including analyzing the frequency spectrum, the energy distribution map, and the signal waveform. Thus, it takes lots of time because all the steps are part of a manual selection. For the proposed method, after the model is well trained, we just need to feed the target signal into the model for the model to output the denoised signal almost immediately. In addition, the well-trained model can be used for any other subjects. Thus, the automagical model should reduce much processing time compared to manual methods.

This experiment consequently proves that GAN can replace the artificial denoising method. After training, it can filter new noise signals and achieve the same performance as the artificial filtering method. The experimental simulations are implemented by using MATLAB R2020b on a Windows personal computer with Core i7-8665U 2.11 GHz CPU and RAM 16.00 GB. The mean denoising execution time for each trial is 0.039 s. In practical applications, if the target dataset is large, some effective filtering methods should be selected first to filter some of the EEG signals, and then the GAN network can be trained to filter other signals automatically. This saves time and achieves the effect of combining multiple methods.

## 5. Conclusions

In this paper, we proposed a GAN-based denoising method. For most noise effects, the GAN denoising model can obtain good filtering performance levels. The correlation between original signal and filtered signal is 0.7771, and the RMSE is 0.0757 when the proposed GAN model is used. Because it is difficult to generate the EEG signal using a generative model, we also introduced a sample entropy and energy-based normalization method which can process the EEG signal before feeding into the model. This can help the generator to produce more real EEG signals. Once the denoising system is established, new subject data can still be filtered even though the EEG signals from this new subject were never involved in the training of the model. Compared to the artificial denoising method, the proposed GAN-based denoising model can achieve similar performance levels compared to artificial denoising method and can make the whole process automatic, which saves much time. Accordingly, the GAN-based denoising method can be used for real-time and online EEG analysis.

## Figures and Tables

**Figure 1 sensors-22-01750-f001:**
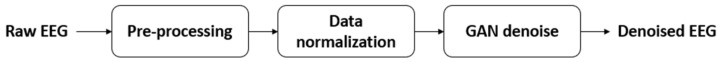
Block diagram of the proposed GAN-based denoising system.

**Figure 2 sensors-22-01750-f002:**
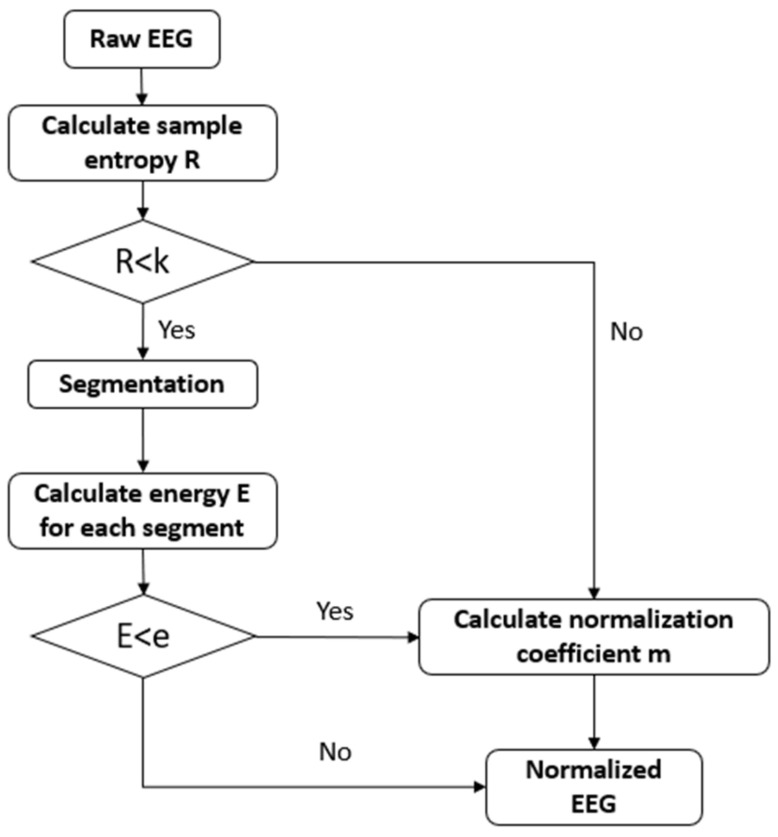
Entropy and energy-based normalization.

**Figure 3 sensors-22-01750-f003:**
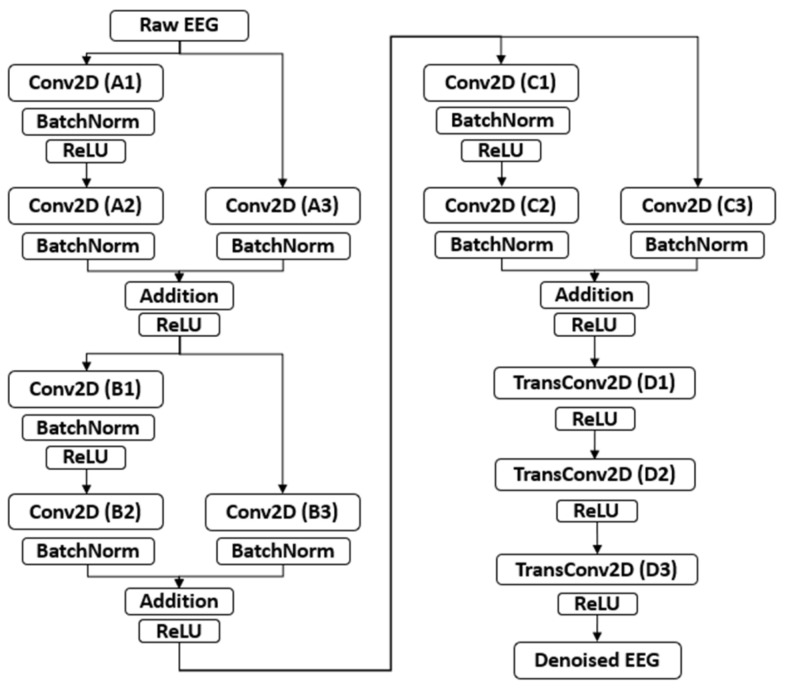
Structure of the generator (Conv2D is the 2-D convolution layer, BatchNorm is the batch normalization layer, and TransConv2D is the transposed 2-D convolution layer).

**Figure 4 sensors-22-01750-f004:**
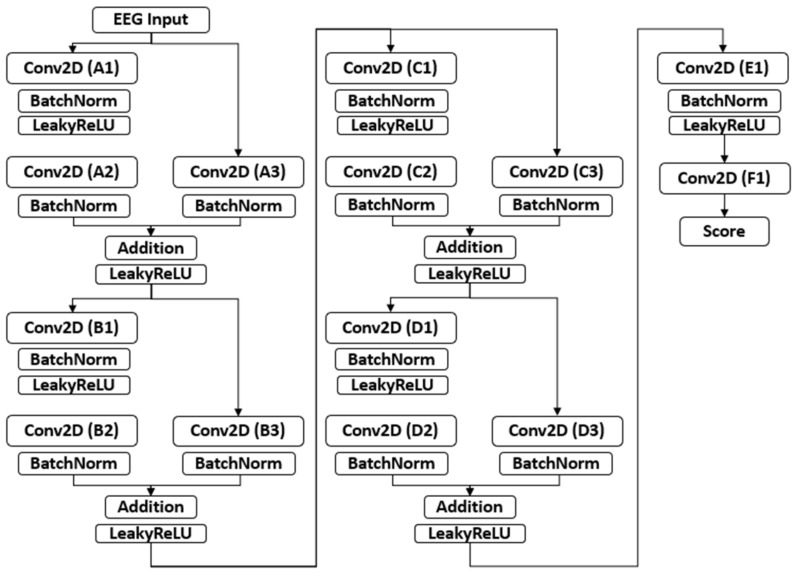
Structure of the discriminator (Conv2D is the 2-D convolution layer and BatchNorm is the batch normalization layer).

**Figure 5 sensors-22-01750-f005:**
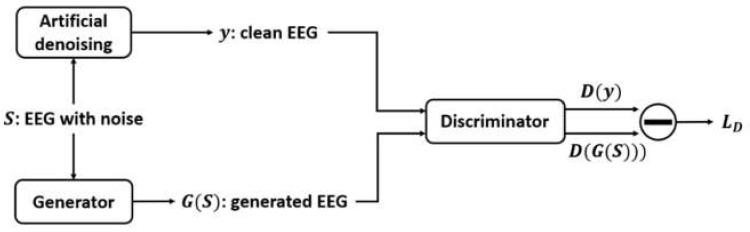
Training process for the discriminator.

**Figure 6 sensors-22-01750-f006:**
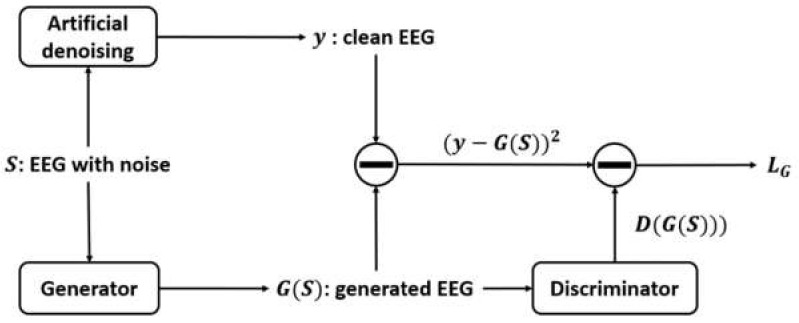
Training process for the generator.

**Figure 7 sensors-22-01750-f007:**
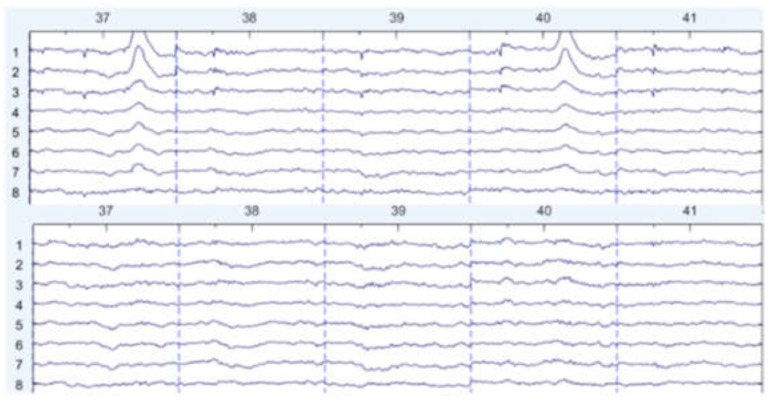
Slightly affected by noise (noisy data at the **top**, filtered data at the **bottom**).

**Figure 8 sensors-22-01750-f008:**
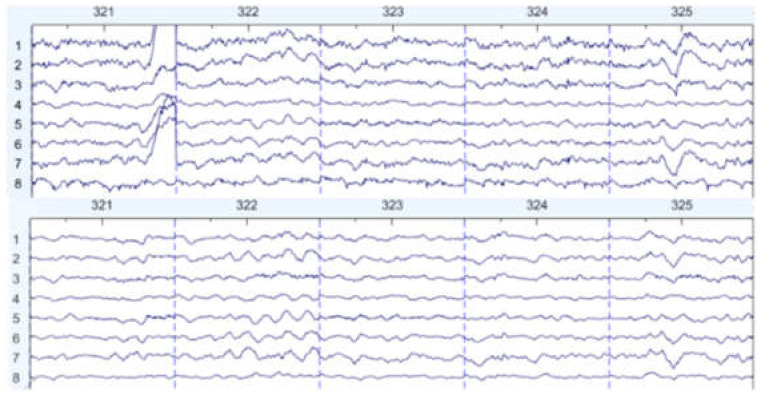
Moderately affected by noise (noisy data at the **top**, filtered data at the **bottom**).

**Figure 9 sensors-22-01750-f009:**
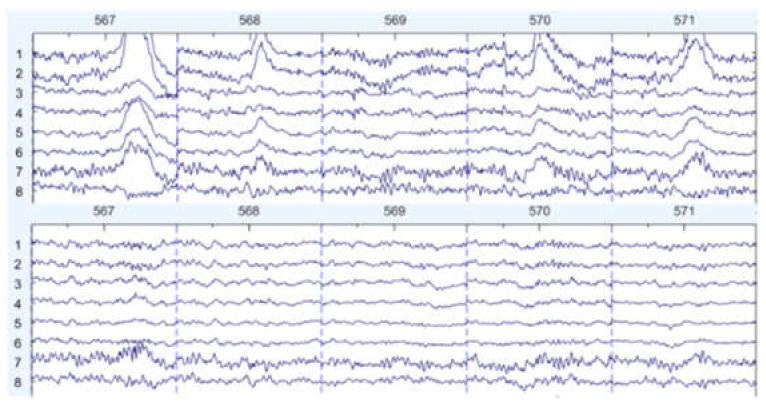
Seriously affected by noise (noisy data at the **top**, filtered data at the **bottom**).

**Figure 10 sensors-22-01750-f010:**
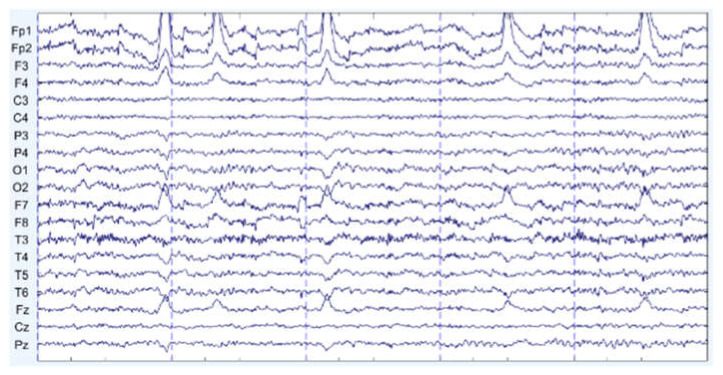
A group of EEG signal with eye movement noise.

**Figure 11 sensors-22-01750-f011:**
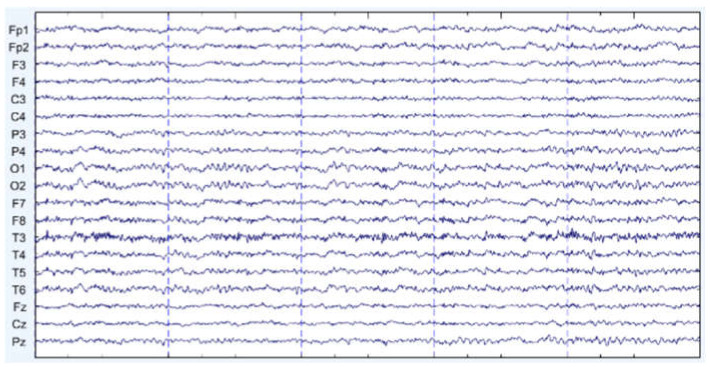
Denoised EEG signal with eye movement noise.

**Figure 12 sensors-22-01750-f012:**
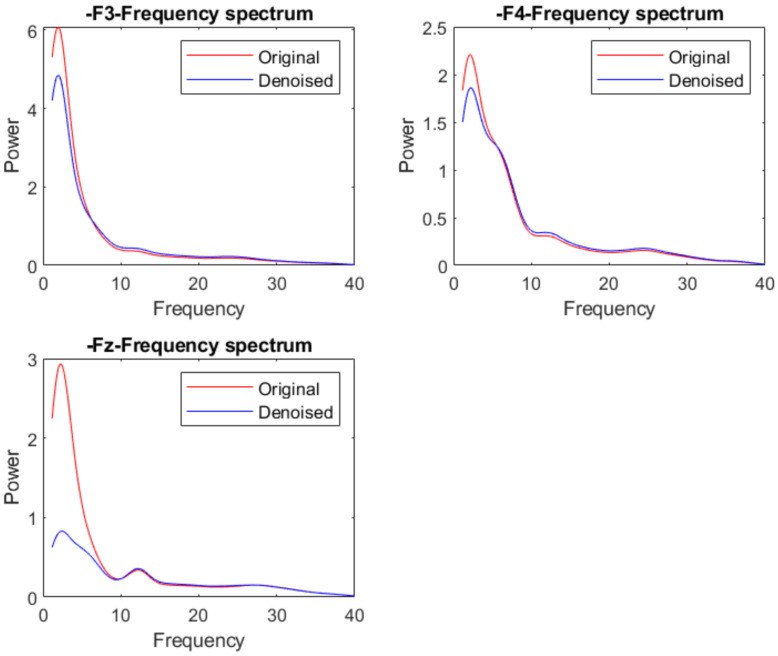
Frequency power spectrum comparison of the signal with eye movement noise and denoised signal.

**Figure 13 sensors-22-01750-f013:**
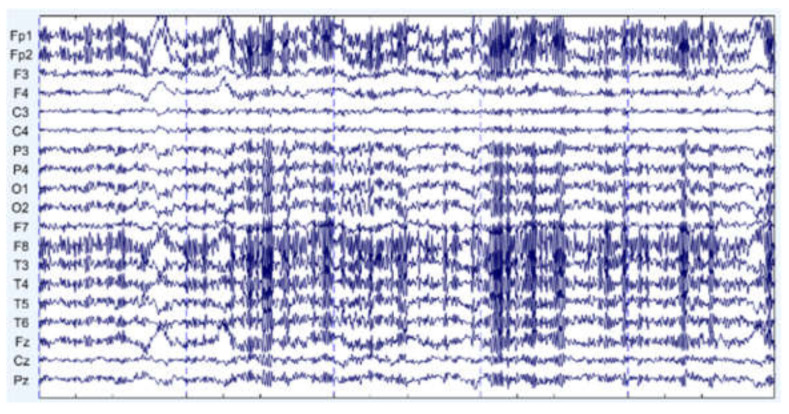
A group of EEG signals with EMG noise.

**Figure 14 sensors-22-01750-f014:**
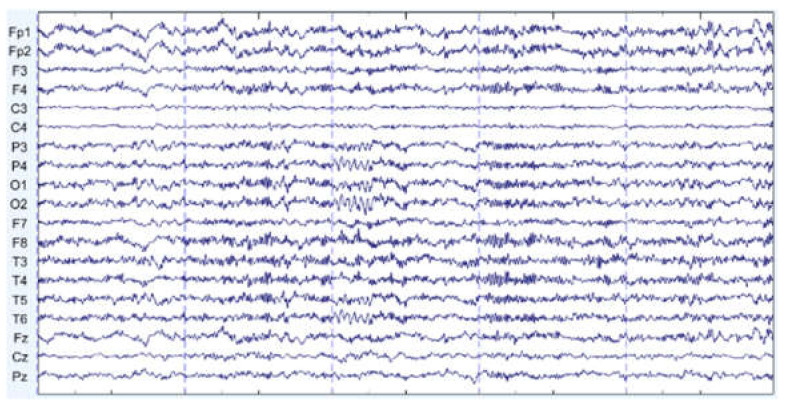
Denoised EEG signal with EMG noise.

**Figure 15 sensors-22-01750-f015:**
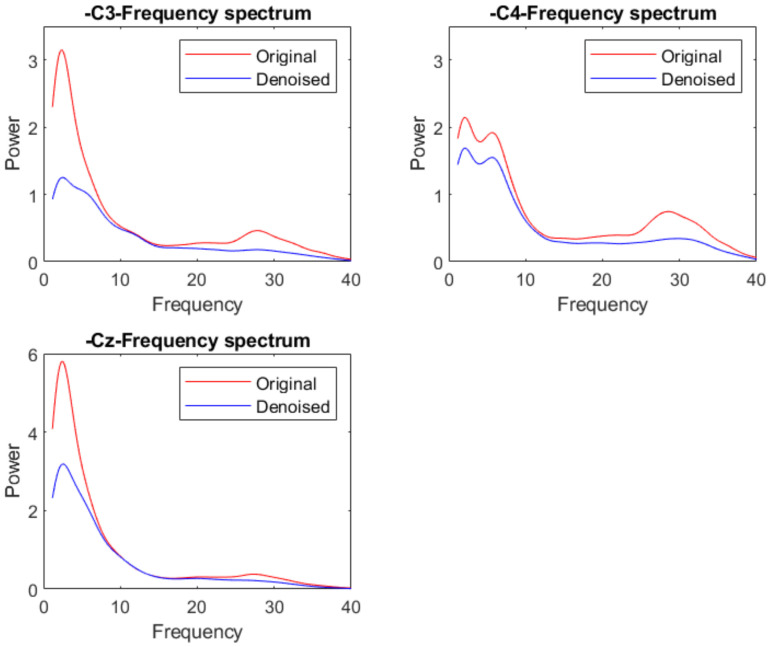
Frequency power spectrum comparison of the signal with EMG noise and denoised signal.

**Table 1 sensors-22-01750-t001:** Generator parameters.

Layer	Filter Size	Stride	Filter Number
A1	[3, 10]	[3, 5]	16
A2	[1, 1]	[1, 1]	32
A3	[1, 1]	[3, 5]	32
B1	[3, 10]	[3, 2]	32
B2	[1, 1]	[1, 1]	64
B3	[1, 1]	[2, 3]	64
C1	[2, 3]	[2, 2]	64
C2	[1, 1]	[1, 1]	128
C3	[1, 1]	[2, 2]	128
D1	[2, 3]	[2, 2]	128
D2	[3, 5]	[3, 2]	64
D3	[3, 10]	[3, 5]	1

**Table 2 sensors-22-01750-t002:** Discriminator parameters.

Layer	Filter Size	Stride	Filter Number
A1	[2, 10]	[2, 4]	16
A2	[1, 1]	[1, 1]	32
A3	[1, 1]	[2, 4]	32
B1	[2, 5]	[2, 2]	32
B2	[1, 1]	[1, 1]	64
B3	[1, 1]	[2, 2]	64
C1	[2, 3]	[2, 2]	64
C2	[1, 1]	[1, 1]	128
C3	[1, 1]	[2, 2]	128
D1	[2, 3]	[2, 2]	128
D2	[1, 1]	[1, 1]	256
D3	[1, 1]	[2, 2]	256
E1	[2, 2]	[2, 2]	512
F1	[1, 4]	-	1

**Table 3 sensors-22-01750-t003:** Number of trials for the HaLT dataset.

HaLT	Trials Number
A	959
B	955
C	959
E	949
F	959
G	952
H	955
I	951
J	946
K	958
L	953
M	960

**Table 4 sensors-22-01750-t004:** Denoising performance using single data.

HaLT	Correlation	RMSE
A	0.8408	0.0672
B	0.8797	0.0629
C	0.6404	0.1019
E	0.7439	0.0802
F	0.8901	0.0616
G	0.5661	0.1176
H	0.7134	0.0869
I	0.7347	0.0884
J	0.8422	0.0732
K	0.8299	0.0667
L	0.7536	0.0845
M	0.8332	0.0596
Mean	0.7723	0.0792

**Table 5 sensors-22-01750-t005:** Denoising performance using merged data.

HaLT	Correlation	RMSE
A	0.8533	0.0667
B	0.8659	0.0649
C	0.7033	0.0838
E	0.7214	0.0793
F	0.8585	0.0641
G	0.6333	0.0958
H	0.7312	0.0858
I	0.7137	0.0891
J	0.8316	0.0673
K	0.8243	0.0682
L	0.7615	0.0825
M	0.8273	0.0614
Mean	0.7771	0.0757

**Table 6 sensors-22-01750-t006:** Comparison of denoising performance.

HaLT	GAN Correlation	GAN RMSE	Artificial Correlation	Artificial RMSE
A	0.8533	0.0667	0.8779	0.0543
B	0.8659	0.0649	0.8429	0.0653
C	0.7033	0.0838	0.6522	0.1050
E	0.7214	0.0793	0.7025	0.0850
F	0.8585	0.0641	0.8768	0.0593
G	0.6333	0.0958	0.5821	0.1180
H	0.7312	0.0858	0.7034	0.0969
I	0.7137	0.0891	0.7547	0.0833
J	0.8316	0.0673	0.8591	0.0622
K	0.8243	0.0682	0.8773	0.0562
L	0.7615	0.0825	0.7848	0.0896
M	0.8273	0.0614	0.8037	0.0625
Mean	0.7771	0.0757	0.7764	0.0781

## Data Availability

The data presented in this study are openly available at https://figshare.com/collections/A_large_electroencephalographic_motor_imagery_dataset_for_electroencephalographic_brain_computer_interfaces/3917698 (accessed on 19 January 2022), reference number [28].

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
