# Peer review of "Auto-Denoising for EEG Signals Using Generative Adversarial Network"

_sensors, 2022, doi:10.3390/s22051750_

Round 1
Reviewer 1 Report
The authors propose a technique to denoise EEG recordings in order to enhance the signal and suppress the artifacts based on GAN networks. In particular their proposal is able to clean EEG data that are contaminated by power line interference, eye blinks as well as muscle activity. Experiments were performed using a database that has EEG recordings during imaginary behavioural tasks (BCI) and the proposed network architecture was tested against traditional artifact and noise removal techniques such as ICA, Wavelets and neural networks.
Reviewer's Comments
=====================
lines 89-91 This is not always the case. There exist ICA based methods that can reject artifacts automatically, like for instance the ICLabel technique (included in the EEGLAB toolkit) that can classify independent components of EEG data into different noise / brain related sources.
lines 96-97 the authors could comment in more detail their phrase "..it performs well when the testing data of subjects are different from the training subjects" and give some references on similar work.
line 102 there is no term such as "brain muscles" that the authors use in the manuscript
line 109-112 is the sample entropy a novel metric that the authors propose to use for the first time, or has it been used in other EEG analysis related problems in the past? If so, the authors should provide more references in the subject.
lines 182-190 the threshold values that the authors propose, are these dependent on the specific problem / EEG recorder / task etc? How can this be generalized to more general cases? Are these dependent on the specific subject or did the authors find small deviations from one subject to the other? Can they include more info on this?
line 205-211 what are the clean data that are used in the training phase? How do you select those trials? Is this done by visual inspection or with another traditional method? Are the data taken from a specific task period like imaginary movement, or from a resting phase?
line 325-326 what specific task do the subjects perform during the 1 sec (200 samples) taken for each trial? Is this exactly after the prompt for an imaginary task or is it at a later stage? This isn't clear from the paper.
line 350-354 the use of correlation between the initial (unfiltered) EEG and the de-noised version, is a very naive metric to measure the efficacy of the proposed denoising method. The authors should try to justify more the use of this criterion as well as the RMSE.
Overall how could the proposed method be compared with traditional methods (such as ICA) when they are used for denoising? Which method needs more data to learn? How is the proposed method compared with the traditional ones in terms of training time?
Do the authors present a denoising method to be specifically applied in the case of BCI experiments? In my opinion this method is very restrictive in the situation that can be used as the experimental protocol has to be defined. I find it very difficult for this approach to give good results in general cases as it wouldn't be so easy to estimate the different values of all the different thresholds that the method uses in other situations which involve normal brain activity.
Finally, to clearly justify the performance of the proposed GAN denoising network compared to the traditional ones, instead of using the correlation metric, it would be more clarifying for the authors to show an improvement of the bci problem when recordings were cleaned with the GAN-based method instead of other methods already proposed in the literature.
Author Response
Thanks the comment, please see the attached response letter.

Reviewer 2 Report
In this paper author presented automatic denoising model for EEG signal using entropy threshold, energy threshold, and GAN. Author describe the paper well However author need to focus on the following issues:
- Author should re-write the abstract to express the contribution of this research.
- Author mention "Artificial denoising Method". This term is not clear enough.
- Author claim the "processing time reduce". There is no experiment to prove that.
- Is this GAN is now? I don't think so. Author should explain the contribution.
- For entropy threshold, k is defined by experience. Is that really reliable?
- In line "The raw EEG data is filtered by 1 to 40 Hz because the main features of EEG exist in this frequency band". Author should provide reference as a proof. Also line 193.
- It did not explain well that how 1D signal put to the 2D conv network.
- Author mentioned "extracts the main features of the signal" at line 216. What are the feature? author did not discussed.
- Two CNN networks are involved. Author should express the result for execution time of denoising.
- Author should improve the writing.
Author Response
Thanks for the comments, please see the attached response letter.

Reviewer 3 Report
In this work, a Generative Adversarial Network (GAN) based denoising method is proposed to denoise the multichannel EEG signals. Results shows that the proposed automatic GANdenoising network achieves the same performance as the manual hybrid artificial denoising method. However, it is claimed that the GAN network makes the denoising process automatic, representing a significant reduction in time. This work is acceptable after following addressing following revisions/queries.
- Why GAN model is preferred for denoising? Why not auto-encoders?
- sample entropy and energy threshold-based data normalization method is introduced in the work. why these methods are used? why data normalization is required for denoising?
- what is blind denoising method? why it is used here?
- give equations for correlation and RMSE. what is the significance of providing correlation here?
- what is eye movement noise? why only this noise is tested?
- Figure 12. Frequency spectrum of denoised signal. change caption.
- Revise the conclusion. Provide your key findings , results here.
Author Response

(The authors gave the same response as above.)

Round 2
Reviewer 1 Report
The authors have taken into consideration all the reviewer's comments and have made all the appropriate additions / corrections in the manuscript. Therefore, it is my opinion that he manuscript can be considered for publication.
One final comment, the authors should include a reference to the EEGLab toolkit (line 222).
Author Response
A reference [24] in line 238 is included to the EEGLab toolkit
Reviewer 2 Report
Author should address the appropriate response of question 3, 4, and 5 in revised manuscript.
Author Response
Thanks for the comments!
Response 3 is added in the revised paper (page 5, line211-226)
Responses 4 and 5 are added in the revised paper (page 17, line 512-523 and line 530-536.